# Fluorescence Spectroscopy for the Diagnosis of Endometritis in the Mare

**DOI:** 10.3390/ani12091157

**Published:** 2022-04-29

**Authors:** Andrea D’Agostino, Tommaso Di Palma, Stefano Cecchini Gualandi, Raffaele Boni

**Affiliations:** 1Department of Sciences, University of Basilicata, Campus Macchia Romana, 85100 Potenza, Italy; andrea.dagostino@studenti.unibas.it (A.D.); stefano.cecchini@unibas.it (S.C.G.); 2Veterinary practitioner, Ruoti, 85056 Potenza, Italy; tao.tao.tdp@gmail.com

**Keywords:** mare, endometritis, ROS detection, spectrofluorometer, uterine fluid turbidity

## Abstract

**Simple Summary:**

A new method is proposed for the diagnosis of equine endometritis, one of the main pathologies affecting the fertility of this species. Cytological diagnosis is commonly based on the microscopic identification of polymorphonuclear cells (PMNs) among the cells recovered from the uterus; when the PMN/endometrial cells ratio exceeds a threshold level, the diagnosis of endometritis may be achieved. PMNs are cells characterized by the content of enzymes capable of producing high quantities of reactive oxygen species (ROS) that are used to neutralize pathogens. Using ROS-sensitive fluorescent labeling, the incidence of ROS in uterine cells has been evaluated with a spectrofluorometer. Following preliminary tests carried out on endometrial cell samples collected from slaughtered animals and mixed with blood leukocytes, calibration lines were obtained for the relative fluorescence intensity. The study was, then, moved to field conditions on mares that were submitted to a uterine flushing before insemination for performing microbiological tests, endometrial cell smears, and spectrofluorometer reading. The here proposed method based on fluorescence spectroscopy demonstrated a high relationship with the microscopic cytological technique, offering easy practicality and speed of execution.

**Abstract:**

By exploiting the PMN property to produce high quantities of oxygen peroxide to neutralize pathogens, the oxygen peroxide content of uterine cells was measured to diagnose endometritis. After preliminary in vitro studies in which endometrial cells from slaughtered mares were mixed with leukocytes from peripheral blood, endometrial samples were collected by uterine flushing from mares before insemination. Staining endometrial cells with H_2_DCF-DA was combined with hydroethidine to normalize the fluorescence intensity with the cellular content of the sample. Stained cell smears were assumed as the gold standard of endometritis, and based on this assay, the samples were considered positive (C+) and negative (C−) for endometritis. The amount and the turbidity of fluid recovered by uterine flushing were significantly (*p* < 0.01) higher in C+ than in C−. Moreover, the oxygen peroxide content of the endometrial cells was significantly higher in the C+ than in the C− group (6.31 ± 1.92 vs. 3.12 ± 1.26, *p* = 0.001). Using the value of 4.4 as the cutoff level of this fluorescence cytology assay, it was found that only one C− sample exceeded the cutoff level (false positives = 7.7%) while three C+ samples showed values below the cutoff level (false negative = 11.5%).

## 1. Introduction

Mare endometritis is a very widespread disease resulting in a lowering of fertility [1,2]. The occurrence of this pathology is usually related to the colonization of the uterine mucosa by pathogens, either bacteria or fungi. In its determinism, however, some hazard determinants such as reduced uterine defense, poor perineal conformation, a uterine pending position, and delayed uterine clearance due to suboptimal myometrial contractility may also be involved [3] as well as individual predispositions. Exploiting experimental induction of endometritis through the uterine infusion of an intrauterine bacterial challenge with *Streptococcus equi zooepidemicus* allowed for discriminating between endometritis-susceptible and -resistant mares [4]. A critical moment for the establishment of this pathology is represented by the insemination intervention, whether natural or assisted. The maneuvers of introducing semen into the female genital tract as well the semen itself evoke an inflammatory response of the uterus [5]. In addition to an ascending route, the bacterial/fungal component underlying the inflammatory process may also derive from the activation of microorganisms already present in the uterus in the dormant form [6]. Post-breeding endometritis is a transient event whose occurrence peaks at 6–12 h and usually disappeared at 48 h post-insemination [5,7]. If the insemination is successful, after five days, the embryo will reach a suitable uterine environment for its development. Alternatively, this inflammatory reaction can continue and grow to create a loop that shifts the uterine reactivity from a physiological to a pathological status [8]. Post-breeding persistent endometritis, bacterial and other infective endometritis, as well as poor uterine clearance represent the etiological bases of endometritis. This pathological condition is stated by the infiltration of inflammatory cells within the uterine mucosa undergoing consequent degenerative phenomena [8]. The diagnosis of endometritis can be achieved through various techniques, such as transrectal palpation of the uterus, ultrasound assessment, vaginoscopy examination, microbiological analysis, or cytological and histological tests based on evaluating uterine cells collected by double-guarded swab, cytobrush, uterine flushing or biopsy. Comparative evaluation studies between these methods failed to reach a definitive result. Using combined criteria for endometritis diagnosis, De Amorim et al. [9] found endometrial biopsy as the most sensitive diagnostic tool to detect equine endometritis whereas abnormal clinical findings and positive cytology showed only moderate sensitivity. On the other hand, the uterine biopsy is an unwelcome procedure to practitioners due to its invasive nature and is accused of providing conflicting results concerning the sampling site; for this, the use of multiple samplings was recommended [10,11]. However, cytological and histological examinations between different uterine biopsy loci of uteri collected at slaughterhouse did not reveal significant differences between specimens [12]. Moreover, in the endometrium, the number of polymorphonuclear leukocytes (PMN) at cytological examination showed fair agreement with the occurrence of PMN in the stratum compactum at the histological examination [12].

Low volume lavage is reputed by several authors as the most sensitive tool to collect uterine samples for the diagnosis of bacterial endometritis [13]. Bacterial infections are usually associated with PMN endometrial infiltration; however, some microorganisms are associated with only heavy debris on cytological specimens [14]. In addition, dormant bacteria, such as *S. zooepidemicus*, residing within the endometrium does not evoke any cytological reactiveness [6].

The cytological analysis is generally based on a cell smear that is colored and analyzed with a microscope. Endometritis diagnostics are obtained based on either PMNs absolute finding or their relative amount over the uterine cells displayed. Each method used suggests a threshold level of PMNs above which the condition of endometritis is recognized (for review see [15]). Riddle et al. [16] reported that endometritis can be defined by the presence of 2–5 PMNs per 400× microscope field, and/or a bacterial monoculture; this occurrence, or exceeding these levels, is associated with reduced pregnancy rates. However, most of the authors agree that the PMN threshold level ranges from 0.5 to 1% of PMNs on the total number of cells recovered [15].

PMNs are leukocytes that in the presence of inflammation migrate from the bloodstream to the inflamed tissue. They play either direct or indirect defense activities through granule proteins that enhance the adhesion of monocytes to the endothelium and stimulate subsequent extravasation of inflammatory monocytes. Moreover, PMN granule proteins activate macrophages to produce and release cytokines and to phagocytose IgG-opsonized bacteria. Furthermore, by direct cell–cell contacts, PMNs activate monocyte-derived dendritic cells enhancing antigen presentation (for review see [17]). Phagocytosis undoubtedly represents a primary defense activity of PMN. The PMN cytoplasm is rich in glycogen and granules with enzymatic content. Myeloperoxidase is released from granules during phagocytosis and is involved in generating hydrogen peroxide (H_2_O_2_), a critical antimicrobial substance [18]. Moreover, NADPH oxidase plays an important role in the generation of H_2_O_2_ and other reactive oxygen species (ROS) [19]. The H_2_O_2_ interacting with myeloperoxidase and a halide, such as chloride, generates the potent antibacterial compound, hypochlorous acid, within the phagosome (for review see [20]). Hence, PMNs produce and contain high amounts of H_2_O_2_. This feature could be exploited for diagnostic purposes, allowing the diagnosis of endometritis to be made through the dosage of H_2_O_2_ in the cells recovered from the uterus.

The present study, therefore, proposes the use of a fluorescent probe for assessing H_2_O_2_ content in cells collected by uterine flushing as an innovative method for the diagnosis of endometritis in the mare.

## 2. Materials and Methods

### 2.1. Reagents

Phosphate buffer saline (PBS, pH 7.4, cell culture tested) was purchased from Gibco (Life technologies, Grand Island, NY, USA). May–Grünwald stain and Gimsa solutions, OmniPure^®^ Water, Collagenase II, Polyvinyl alcohol (PVA), dimethyl sulfoxide (DMSO), and fetal calf serum were purchased from Sigma Chemical Company (Milan, Italy) and cell culture tested. 2′,7′-dichlorodihydrofluorescein diacetate (H_2_DCFDA) and hydroethidine (HE) were obtained from Life Technologies (Milan, Italy). Ringer’s lactate was purchased by SALF spa (Laboratorio Farmacologico S.A.L.F., Bergamo, Italy).

### 2.2. Animals

The study was developed in a preliminary (in vitro) phase concerned the development of the technique (from October 2021 to January 2022) and an in vivo phase where the technique was tested under field conditions (from February to March 2022). The study involved 15 regularly slaughtered yearlings and 17 mares housed on various private farms of the Potenza district (Basilicata, Italy) and routinely subjected to uterine flushing for microbiological tests before undergoing artificial insemination. In the case of a positive microbiological finding, the animals were subjected to antibiotic treatments based on the antibiogram or to treatments with topical aids and subsequently subjected to new uterine flushing. Insemination was carried out only following the negativization of the microbiological test. The animals used for the in vitro phase were of different breeds and from 1 to 2 years old; the mares enrolled in the in vivo test were quarter horses with a mean (±SD) age of 9.6 ± 5.8 years and 1.4 ± 3.0 number of births. The latter animals did not have particular health or reproductive problems detected in the anamnesis or conformational problems that would require special attention; they were client-owned and informed owner consent was obtained.

### 2.3. Ethics Approval

This study has been conducted according to the guidelines of the European Directive 63/2010 on the protection of animals used for scientific purposes, transposed into the Italian law by Legislative Decree 2014/26. Considering that the proposed experimental design does not fall within the European Directive 63/2010, the Ethics Committee of the University of Basilicata (OPBA) has established that it did not require any authorization for its performance.

### 2.4. Collection of Uterine Cells

Uterine cells were collected either from slaughtered or living mares. In the formers, genital tracts were collected on evisceration and transported to the laboratory at 4 °C. The uterine horns were aseptically opened and the mucosa was scratched gently with a scalpel blade. The cells were collected in a conical tube and washed in PBS added with 0.1% PVA (PBS-PVA). After centrifugation at 200 g for 10 min, these cells were enzymatically dissociated by treating with 2 mg collagenase II mL^−1^ at room temperature (RT) under a continuous vortex for 4 min. After that, an equal amount of PBS supplemented with 20% FCS was added. The material obtained was filtered (Cell Strainer 40 µm, Fisher Scientific Italia, Rodano, Milan, Italy) to remove cellular aggregates and submitted to double washing with PBS with 10% fetal calf serum (FCS). In the end, the cells were counted with a Bürker chamber and stored at 4 °C up to two hours until use.

The in vivo collection of uterine cells was carried out by uterine flushing [21]. Briefly, mares were restrained in stocks, and their perineal regions were carefully cleaned and disinfected with povidone–iodine scrub. Then, a uterine catheter was manually introduced through the vagina into the uterus and 500 mL of sterile Ringer’s lactate was infused into the uterus. Subsequently, the Ringer’s lactate flask was positioned at the bottom to recover the infused liquid by gravity flow. The collected samples were stored at 4 °C and transported to the laboratory. There, after removing samples to be used for microbiological and turbidimetric tests, the collected liquid was aseptically divided into 50 mL conical tubes and centrifuged (400 g for 10 min). The pellet from each tube was collected with a small amount of PBS-PVA in a new collection tube. The resulting cell suspension was further centrifuged and resuspended in PBS-PVA. Cells samples were stored in PBS with 10% FCS at 4 °C up to two hours until use.

### 2.5. Turbidity Evaluation of the Collected Uterine Fluid

A 4 mL sample of each fluid collected by uterine flushing was transferred in a disposable cuvette and read with a spectrometer. In the preliminary phase of the study, fluid samples recovered from uterine flushings were scanned in the absorbance spectrum between 380 and 700 nm. Although a clear specific absorbance peak was not identified, an absorbance value of 550 nm was chosen for subsequent readings, similarly to Mattos et al. [22]. At each reading, the solution used for uterine lavage was used as a blank.

### 2.6. Leukocyte Isolation

Blood samples were collected from the jugular vein into a 10 mL vacutainer containing sodium heparin that was gently moved upside down several times to mix the heparin and prevent coagulation. Blood was first centrifuged (500 g for 10 min) and 1 mL plasma and possible buffy coats were aseptically aspirated and discarded [23]. Red blood cells were lysed by osmotic shock [24]. Briefly, 2.5 mL of blood were mixed with 5 mL of OmniPure^®^ water and, then, osmolarity was restored with 2.5 mL of 2.7% NaCl solution. Samples were twice washed with PBS supplemented with 10% FCS by centrifugation at 200 g for 10 min. Finally, leukocytes were resuspended in PBS + 10% FCS, counted with a Bürker chamber, and stored at 4 °C up to two hours until use. In agreement with previous studies [25], we estimated that the leukocytes isolated from the blood contain approximately 49–53% of PMN.

### 2.7. Cytological Test

Cytological smears were obtained by placing 4 µL of a concentrated cell suspension on a slide, previously cleaned with methanol, and dispersing the liquid with the tip of the pipette within a circular space. Samples were air-dried for a few minutes before they were fixed and stained with May–Grünwald stain solution (0.25% (*w/v*) in methanol). After 3 min, distilled water at the same amount used for the dye was deposited on the slide. After a further 3 min, having removed the liquid on the slide, a Gimsa solution (1:10 diluted in distilled water) was placed on the slide and left for 15 min. At the end, the slide was washed with tap water, air-dried and observed at the microscope [26]. In the present study, this test was used as a diagnostic gold standard for endometritis.

### 2.8. Cell Viability Evaluation

Trypan Blue (TB) viability measurement was performed by standard methods [27]. TB solution (0.4% wt/vol, GIBCO) was mixed 1:1 with each cell sample. In detail, a 4 μL drop of stain was placed on a microscope slide, mixed with 4 μL of cell suspension, and covered with a coverslip. Cells were evaluated with a Nikon light microscope at 400×; cells that stained blue were scored as nonviable; cells that did not load the dye were scored as viable. Cell viability was calculated on the basis of the number of unstained cells out of the total number of cells counted (stained + unstained = 400 cells per sample).

### 2.9. Fluorescent Targets and Spectrofluorimetric Analysis

Intracellular H_2_O_2_ levels were evaluated by 2′,7′-dichlorodihydrofluorescein diacetate (H_2_DCF-DA), whereas hydroethidine (HE) was used to label cell nuclei to normalize ROS levels based on the sample cell numbers.

H_2_DCF-DA is a cell-permeable dye; in the cell, it is cleaved by esterase producing the nonpermeant nonfluorescent 2′,7′-dichlorodihydrofluorescein (H_2_DCF). The H_2_DCF is oxidized by H_2_O_2_ to the highly fluorescent 2′,7′-dichlorofluorescein (DCF). It is excited at 490 nm wavelength and emits fluorescence at ~520 nm wavelength. H_2_DCF-DA stock solution was prepared in DMSO at 80 mM and stored at −80 °C.

HE is the sodium borohydride-reduced form of ethidium. It is a cell-permeable dye showing a blue fluorescence in the cytoplasm until oxidized; when oxidized, it generates two red fluorescent products, i.e., the 2-hydroxyethidium (2OH-E) and the ethidium (E+). The former is a specific indicator of superoxide anions (O_2_^−^); the latter binds to DNA and emits fluorescence at ~600 nm wavelength in response to 490-nm wavelength excitation. HE stock solution was prepared in DMSO at 3 mM and stored at −80 °C.

Cell samples were suspended in 200 µL PBS-PVA and treated with 800 µM H_2_DCF-DA. After 30 min incubation, cell suspensions were centrifuged for 5 min at 200 g, the pellet was resuspended in PBS-PVA and incubated for an additional 20 min. After another centrifugation (5 min at 200 g), cells were suspended in PBS-PVA and treated with 30 µM HE. After 30 min, samples were transferred to the quartz cuvette for spectrofluorometric analysis. Fluorescence intensity (arbitrary units, AU) was analyzed in the 500–650 nm emission wavelength spectrum following a 490 nm wavelength excitation.

### 2.10. Microbiological Analyses

Uterine flushing samples were inoculated in enrichment broth (Brain–Heart infusion) and incubated at 37 °C [28]. After 12–24 h, sowing was carried out on MacConkey and BD Columbia CNA agar plates enriched with 5% sheep blood and incubated at 37 °C for the isolation of Gram negative and positive bacteria, respectively. The inoculated plates were inspected for bacterial growth after 24 and 48 h. In case of growth of bacterial colonies in the MacConckey plates, single colonies were collected, diluted in saline solution, and sown on Mueller–Hinton agar plates for the preparation of an antibiogram. In the case of colony growth in the BD Columbia CNA, single colonies were harvested and seeded on blood agar plates to test for microbial hemolytic potential and to achieve germ purification. Mannitol salt agar was used to selectively isolate *Staphilococcus aureus*. Microbial characterization was performed according to cultural, morphological, and biochemical properties according to standard laboratory methods [29,30].

### 2.11. Experimental Design

In the in vitro phase, numerous preliminary tests were conducted to identify (i) the fluorochrome that best highlights the presence of PMN; (ii) the fluorochrome that best allows a reliable normalization of the absolute value of the fluorescence intensity in relation to the cell number; (iii) the minimum quantity of cells for obtaining repeatable and solid results. In addition,

(1)increasing number of either uterine cells or leukocytes were loaded with H_2_DCF-DA in order to evaluate the basal content of H_2_O_2_ in these cell populations;(2)increasing numbers of uterine cells loaded with either H_2_DCF-DA or HE as well as in combination have been examined in order to verify the variations in the fluorescence intensity of these fluorochromes at cell number increasing either when singly used or when combined;(3)standard quantity of uterine cells (1 × 10^6^) was mixed with an increasing number of leukocytes (0.5, 1.0, 1.5, and 2%) and loaded with H_2_DCF-DA and HE in order to evaluate the sensitivity of this method of detecting variable quotas of leukocytes inside the uterine cells.

In the second part of the study (in vivo phase), all the fluid collected by uterine flushing in the mares before artificial insemination were transported to the lab in a thermal tank at +4 °C. There, an aliquot of each sample (5 mL) was used for microbiological analysis; another 4 mL aliquot was used for testing the fluid turbidity; finally, after measuring the fluid volume, the remaining amount of the liquid was centrifuged, the uterine cells were, then, isolated and used for cell smearing and fluorescent spectroscopy.

### 2.12. Statistical Analysis

Experimental data were entered into a datasheet and were analyzed by Systat 11.0 (SYSTAT Software Inc., San Jose, CA, USA). The Shapiro–Wilks test was used to evaluate the normal data distribution and Levene’s test to evaluate the homogeneity assumption needed for carrying out parametric tests. Variables displaying a not normal distribution, as percentages, were transformed into angles corresponding to arcsine of the square root for variance analyses. Analysis of variance (ANOVA) was used to test the difference between negative and positive samples at the cytological evaluation in volume and turbidity of the recovered fluid as well as in concentration and H_2_O_2_ content of the cells isolated from such fluid. Coefficients of correlation (R) were calculated by linear regression procedure. The minimum level of statistical significance was *p* < 0.05. Values are presented as mean ± standard deviation (SD).

## 3. Results

### 3.1. In vitro Study

#### 3.1.1. H_2_O_2_ Content in PMN vs. Endometrial Cells

Figure 1 shows the fluorescence intensity (arbitrary units) of 0.1, 0.25, 0.5, and 1 × 10^6^ endometrial cells (A1) as well as 5, 50, and 500 × 10^3^ leukocytes (A2) loaded with H_2_DCF-DA and read with the spectrofluorometer. The fluorescence intensity increased progressively together with the cell numbers, with a highly significant (*p* < 0.001) correlation coefficient for both (endometrial cells R = 0.993 and leukocytes R = 0.996). Comparing the values obtained with the same cell concentration (500 × 10^3^/mL), the leukocytes showed a H_2_O_2_ content about eight-fold higher (390 ± 15 vs. 48 ± 2) than the endometrial cells. A different fluorescence intensity between these two cell populations is also shown in the images obtained with the fluorescence microscope (Figure 1B1, B2).

#### 3.1.2. Single or Combined Use of H_2_DCF-DA and HE in Endometrial Cells

Figure 2 shows the fluorescence spectra related to H_2_DCF-DA and HE in endometrial cells when used single or in combination. The fluorescence intensity increased linearly together with the number of cells in the sample. Significant correlations emerged both between the number of cells of the sample and the fluorescence intensity of DCF (R = 0.993, *p* = 0.01) and E+ (R = 0.934, *p* = 0.01) as well as between DCF and E+ (R = 0.936, *p* = 0.01). Comparing the fluorescence intensity peaks obtained as a result of loading the fluorochromes individually or in combination, no differences emerged. Hence, the combination of these two dyes did not create interference, and the fluorescence intensity peaks of both dyes progressively varied in relation to the cell number, accordingly.

#### 3.1.3. Detection of H_2_O_2_ in Increasing Leukocyte Numbers within Endometrial Cell Samples

Figure 3 shows the fluorescence spectra of 1 × 10^6^ uterine cells mixed with an increasing number of leukocytes, (0.5, 1.0, 1.5 and 2%). The fluorescence intensity of DCF increased progressively as the concentration of leukocytes increased. However, the spectra obtained from samples mixed with 0.5 and 1% of leukocytes overlap, highlighting the sensitivity limit of the method.

### 3.2. In Vivo Study

A total of 39 samples of uterine flushing fluid samples were collected from 17 mares in the estrus phase before attending artificial insemination, and they were analyzed. The uterine flushing procedure was performed only once in ten of these subjects while the remaining seven, because they were affected by endometritis, were treated with topical treatments (antibiotics and disinfectants) followed by a repeated number of uterine flushings, which ranged from 2 to 10. Using the cytological test result as a discriminating parameter, the analyzed samples were divided into two groups (Group C−, *n* = 13, and Group C+, *n* = 26). C+ showed a higher volume (682 ± 193 vs. 419 ± 141 mL; *p* < 0.001) and turbidity (0.469 ± 0.476 vs. 0.014 ± 0.011 abs_550_, *p* = 0.013) of the liquid recovered than C− samples (Table 1). Cell concentration (2527 ± 6350 vs. 9 ± 17 × 10^3^ cells/mL, *p* = 0.133) and viability (54.0 ± 27.7 vs. 71.0 ± 1.7%, *p* = 0.431), although significantly higher and lower in the C+ compared to the C− group, respectively, did not show statistically significant differences due to the high variability of the results (Table 1). Moreover, the H_2_O_2_ content of the endometrial cells was significantly higher in the C+ than in the C− group (6.31 ± 1.92 vs. 3.12 ± 1.26, *p* = 0.001) (Table 1). In order to evaluate the positivity or negativity of the sample based on the fluorescence cytology assay here proposed, a cut-off level was created by summing the mean with the standard deviation of the normalized fluorescence values of the samples estimated to be negative on the cytology test with stained cell smears. Using the value of 4.4 as the cut-off level of the spectrofluorimetric test, it was found that only one C− sample exceeded the cut-off level (false positives = 1/13 = 7.7%) while three C+ samples showed values below the cut-off level (false negatives = 3/26 = 11.5%) and two values fell on the cut-off value (Figure 4).

The microbiological examination showed a positivity for *Escherichia coli* in 46.2% (18/39), for *Klebsiella* spp. in 5.1% (2/39), for *Streptococcus* spp. in 2.6% (1/39), for *Staphylococcus aureus* in 2.6% (1/39), for *E. coli* + *Klebsiella* spp. in 5.1% (2/39), for *E. coli* + *Streptococcus* spp. in 2.6% (1/39), and for *Klebsiella* spp. + *Streptococcus* spp. in 2.6% (1/39) of the samples. The smear cytological test was in agreement with the microbiological test in 94.9% (37/39) of cases. The two samples that did not agree were represented by a sample positive on the cytological but negative on the bacteriological test (a) and a sample negative on the cytological but positive on the bacteriological test (b). The latter was the only false positive sample on the fluorescence cytological assay (b = 6.7) while the first was also positive on the fluorescence cytological assay (a = 6.3). Using the same approach for the other two parameters showing significant differences between C+ and C− samples, we found seven false positives (53.8%) and three false negatives (11.5%) for the volume of the fluid collected at uterine flushing (cut-off level = 560 mL), whereas for the turbidity of such fluid (cut-off level = 0.025 Abs_550_), the number of false positives was two (15.4%) and the number of false negatives was two (11.5%). All these cases concerned samples in which there was no discrepancy between the data obtained following the cytological and bacteriological tests.

## 4. Discussion

A new cytological assay for the diagnosis of mare endometritis is here proposed and evaluated by either simulation tests carried out on cells collected at the slaughterhouse or in-field conditions on cells collected by uterine flushing. Based on the remarkable ability of PMNs to produce H_2_O_2_ for their microbicidal activities, we developed a method that uses a fluorochrome for detecting H_2_O_2_ and a spectrofluorometer for reading H_2_O_2_ levels. To normalize the H_2_O_2_ level on the number of endometrial cells in the sample, a second fluorochrome has been used for evaluating the quantity of DNA, and, therefore, indirectly, the number of nuclei/cells contained in the sample. By combining these two fluorochromes and using a single excitation wavelength, a spectrum with two peaks at about 520 and 590 nm of emission was obtained, allowing a ratiometric measurement of H_2_O_2_ to be used for the diagnosis of endometritis in the mare.

The use of fluorescent probes for the evaluation of some cellular proprieties and activities is widespread and, generally, based on microscopic or flow cytometry assessments. The microscopic investigation is accurate but generally based on a limited number of cells. Flow cytometry is undoubtedly a more powerful and sophisticated method that requires, however, very expensive equipment, high professionalism to use, and perfectly dissociated cells [31]. Studies that have used this technique to evaluate aspects related to endometritis have been carried out in humans and cattle for pathogenetic investigations on the mechanisms related to endometritis occurrence. In humans, dissociating cells from uterine biopsies, abnormal endometrial lymphocyte subpopulations were found in infertile women with chronic endometritis [32]. In cattle, cells collected from either peripheral blood or uterine flushing were characterized on selected immunity parameters highlighting immunity dysfunctions in endometritis occurrence [33]. Fluorescence spectrometry may represent a valid alternative to the above techniques because it is easy to use and capable of providing solid and repeatable results on multiple functional characteristics of the cells highlighted by the use of specific fluorochromes. This methodology has been used for cellular analysis for evaluating sperm quality parameters [34,35] or granulosa cell features for indirect assessment of either follicle or oocyte quality [36]. Weaknesses of this methodology may be attributable to the difficulty of excluding possible contaminants in the sample, such as cell debris and other components capable of providing nonspecific fluorescence as well as aggregates and cellular matrices potentially capable of retaining the fluorochrome and magnifying, unspecifically, the fluorescence intensity of the sample. However, by associating, in the preliminary phase, this analysis with microscopic examinations of the fluorescent target and adopting appropriate positive controls, it is possible to reduce the extent of this source of error and make the technique solid from an analytical point of view.

The analyses conducted in the present study demonstrated that the technique proposed is valid and reliable with repeatable results when applied under both defined conditions, ensured by the combination of endometrial cells and leukocytes, and in-field conditions from cells recovered by uterine flushings. Criticalities were represented by false-positive and false-negative results. For the former, the fact that only one false-positive sample was actually positive to the microbiological test would strengthen the diagnostic value of the proposed method, making it more sensitive than the traditional cytological test used as the gold standard. We do not have a definitive explanation for the false-negative samples. This occurrence may be associated with the different viability of the cell populations collected. Lower cell viability was, in fact, detected in the positive samples at the cytological test. The degenerative or necrotic conditions that inflammatory cells undergo at the end of their function due to autophagy [37,38] or NETosis [39] could significantly reduce their H_2_O_2_ content, bringing it below the cut-off value we have identified.

Scarce information is currently available on the mechanisms of the onset of endometritis in the mare. It could derive from high endometrial sensitivity, peculiar to this animal species and attested by the widely described post-insemination phlogistic phenomena [40,41]. The chronicization of this inflammatory response, as well as bacterial/fungal infections by either ascending route or the virulentation of dormant pathogens, constitute the etiopathogenetic basis of this widespread disease capable of strongly compromising fertility of this species [42]. Particular attention should be paid to the reactivity of endometrial cells towards this inflammation, a condition that can significantly affect one of the main diagnostic tests of endometritis such as the cytological assay. To deepen this aspect, it is necessary to consider the type of cells present in the endometrium (for review see [42]). The cellular composition of the endometrium varies according to the reproductive state of the mare, whether cyclical or pregnant as well as through the estrus cycle [43]. The endometrial epithelium is made up of a layer of luminal, glandular, and staminal cells. Among the luminal cells, there is a variable percentage of ciliated cells. These cells, like those detected in the bronchial epithelium, are characterized by an intense energy consumption for the movement of the flagella, and, therefore, by high metabolism, intense mitochondrial activity with large quantities of ROS produced [44]. Hence, in the method here proposed, the different incidences of these cells within the analyzed sample might compromise the analytical result by interfering with PMN detection and making less clear the difference between mares with and without endometritis.

Preliminary relevant information on the pathophysiological state of the endometrium may rely on the volume and turbidity of the collected uterine fluid. These two parameters may quickly discriminate between physiological and pathological uterine conditions, and in most cases, they are so marked as to be visually appreciable. If the volume of the collected liquid is strongly associated with the presence of liquid in the uterine cavity, representing a diagnostic finding easily obtainable by uterine ultrasonography and widely used in the clinical diagnosis of endometritis [45], the turbidimetric examination represents a very poorly evaluated diagnostic tool that can be easily used for this pathology also in the clinical practice. The accumulation of uterine fluid, especially after breeding, represents a hallmark sign of clinical endometritis and is associated with decreased pregnancy rates [46]; however, the diagnostic sensitivity of this method is low. When compared to cytological examination, this diagnostic method provided a high number of false-positive and false-negative (25.0 and 36.5%, respectively) responses [47]. These values make this finding unreliable from a diagnostic point of view and are confirmed in the present study in reference to the volume of fluid recovered following uterine flushing. On the other hand, an evaluation that could profitably be thoroughly investigated is related to the turbidity of the fluid recovered from uterine flushing. In our study, this test provided excellent evidence of well discrimination between cytology-based positive and negative samples with a low rate of false-positive and false-negative responses. The assessment of the turbidity of the fluid collected following uterine flushing is not a test commonly used for the diagnosis of endometritis in the mare. Up to our knowledge, it was only used to evaluate the therapeutic potential of repeated uterine flushings with saline solution in mares subjected to experimental infection with *Streptococcus zooepidemicus* [22]. Although cell concentration was not a parameter capable of efficiently discriminating between positive and negative endometritis samples, and turbidity is strongly associated with the cellular content of the uterine lavage (R = 0.647; *p* < 0.001, data not shown), the assessment of the turbidity of this liquid provides information that can be easily and quickly obtained and useful information on the uterine state.

In conclusion, the here proposed fluorescence cytological assay for equine endometritis proved to be valid and solid methodology with both in vitro simulation tests and in-field conditions, allowing us to effectively discriminate samples with positive and negative cytological results. The highlighting of variables that may potentially affect the result, such as the content of ciliated cells and the cell viability of the samples, suggesting important elements for further improving the diagnostic sensibility of this technique. The use of fluorochromes to assess the presence of H_2_O_2_ in endometrial cells is an innovative technique that, when properly investigated, can provide important information for the etiopathogenetic characterization as well as diagnostics of endometritis. Currently, cytological analysis is the most reliable and fastest diagnostic technique for endometritis in the mare; however, the use of an alternative and more sensitive technique may be required in those intermediate situations where it is not possible, with traditional methods, to diagnose inflammation. A further diagnostic finding that, although not falling within the primary objectives of this study, has provided interesting results and which deserves an in-deep study is the turbidimetric examination of the uterine flushing fluid. The simplicity and speed of this test could reserve interesting applicative feedback in clinical practice. The present study proposed an alternative approach to the diagnosis of endometritis in mares that not only proved reliable but also allowed us to highlight unexplored aspects of this pathology, opening new diagnostic opportunities.

## Figures and Tables

**Figure 1 animals-12-01157-f001:**
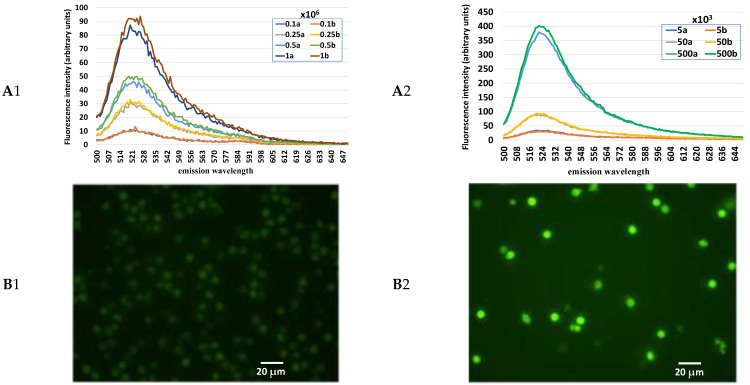
Fluorescence spectra obtained from samples of equine endometrial cells (**A**1) and leukocytes (**A**2) loaded with 2′,7′-dichlorodihydrofluorescein (H_2_DCF-DA) and excited at 490 nm wavelength at increasing cell concentration and in duplicate. Fluorescence images of equine endometrial cells (**B**1) and leukocytes (**B**2) loaded with H_2_DCF-DA.

**Figure 2 animals-12-01157-f002:**
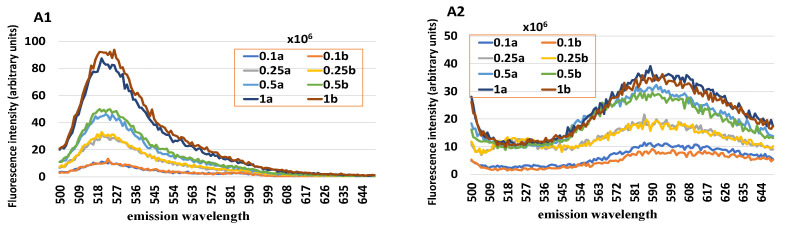
Fluorescence spectra obtained from samples of equine endometrial cells loaded with 2′,7′-dichlorodihydrofluorescein (H_2_DCF-DA) (**A**1), hydroethidine (**A**2), and both the fluorochromes (**A**3), excited at 490 nm wavelength at increasing cell concentration and in duplicate. F_0_A marks the peak of the emission spectrum relative to the DCF (green circle); F_0_B marks the interval of the emission spectrum of the E + used to calculate an average peak value (red square). The normalization of the DCF fluorescence intensity based on the number of cells of the sample is obtained from the F_0_A/F_0_B ratio.

**Figure 3 animals-12-01157-f003:**
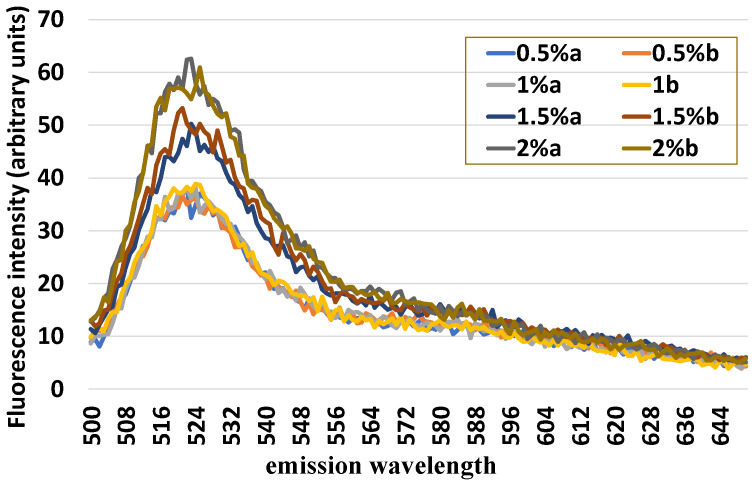
Fluorescence spectra obtained from samples of equine endometrial cells (1 × 10^6^) mixed with increasing (0.5, 1.0, 1.5, and 2%) quotas of leukocytes, in duplicate, loaded with 2′,7′-dichlorodihydrofluorescein (H_2_DCF-DA) and hydroethidine and excited at 490 nm wavelength.

**Figure 4 animals-12-01157-f004:**
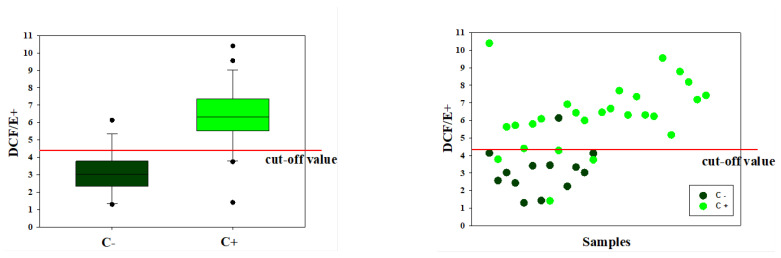
Normalized fluorescence intensity in relation to cell content (DCF/E+) in positive (C+) and negative (C−) cell specimens at the stained smear cytology assay. On the left, values of the medians with error bars at the 10th, 25th, 75th, and 90th percentiles and cut-off value. On the right, the distribution of values in relation to a cut-off value obtained from the mean + SD of the C− group.

**Table 1 animals-12-01157-t001:** Mean (±SD) values obtained from samples of uterine flushing which were positive and negative on the cytological test and related to the quantity and turbidity of the fluid recovered following the infusion of 500 mL of Ringer’s lactate as well as to the concentration, viability and H_2_O_2_ content of the cells isolated from that fluid.

		Positive Samples at the Cytological Test (C+)	Negative Samples at the Cytological Test (C−)	*p*-Value
Samples	*n*	26	13	
Volume	mL	682 ± 193	419 ± 141	0.001
Turbidity	Abs_550_	0.469 ± 0.476	0.014 ± 0.011	0.013
Cell concentration	×10^3^/mL	2527 ± 6350	9 ± 17	0.133
Cell viability	%	54.0 ± 27.7	71.0 ± 1.7	0.431
Intracellular H_2_O_2_ content	U.A.	6.31 ± 1.92	3.12 ± 1.26	0.001

## Data Availability

Data is contained within the article. Detailed data are available on request from the corresponding author.

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
