# Peer review of "Fluorescence Spectroscopy for the Diagnosis of Endometritis in the Mare"

_animals, 2022, doi:10.3390/ani12091157_

Round 1
Reviewer 1 Report
Brief summary:
Authors introduced a new fluorescence cytological assay for equine endometritis diagnostics and proved its validity both in in vitro and in-field conditions. The use of an alternative and more sensitive diagnostic technique was proposed for those stages where cytological analysis as the standard technique was not possible to diagnose uterine inflammation. Fluorescence spectroscopy proved to be reliable and opened new diagnostic opportunities by highlighting previously by traditional methods undetected pathological aspects of endometritis in mare.
Broad comments:
The topic is relevant. The study was properly planned and carried out, the presentation and statistical analysis of the data are appropriate. However, some doubts rose concerning the possibilities of applicability of the described diagnostic method. What is Authors’ experience, does the proposed diagnostic method work in any situation in which endometritis occurs?
Furthermore, considering that fluorescence was performed only when turbidity of the samples applied the absorbance spectrum of 550 nm, would it be possible to also perform the examination using a lower or higher spectrum?
The English language is appropriate, the few typset errors are mentioned and should be corrected.
Specific comments:
Introduction:
L44: „...poor perineal conformation, a uterine pending position...”
Material and Methods:
L110: „®” in upper case
L119: „...involved 15 regularly slaughtered mares and 17 mares housed...”
L133: „...from slaughtered or living mares...”
L140: „…40 µm…”
L162: „…10 mL…”
L164: „…1 mL…”
L166: „…2.5 mL…5 mL…”
L167: „…2.5 mL…”
L172: „...4 µL...”
L200: „…490 nm…”
L218: „…Staphylococcus aureus...”
L251: „…procedure. The minimum…”
L269: „Single or combined…”
L271: „…used single or…”
L298: „…flushings, which…”
L328: in Table 1: „n” without the dot
Discussion:
L359: „...because it is easy to use...”
L401: „…of the collected uterine fluid….”
References:
The following references should be revised: 2, 5, 10, 14, 16, 17, 18, 24, 25, 26, 27, 28, 30, 35, 37, 41, 42.
Typical mistakes: abbreviation of some journal names while full names in others, misspelling, not using initial capital letters in journal names, occasionally missing endpages.
Author Response
Reviewer 1
The topic is relevant. The study was properly planned and carried out, the presentation and statistical analysis of the data are appropriate. However, some doubts rose concerning the possibilities of applicability of the described diagnostic method. What is Authors’ experience, does the proposed diagnostic method work in any situation in which endometritis occurs?
Reply. Thanks a lot, we greatly appreciate your valuable comments. Regarding our experience, of course, we do not know whether this method can be used in any condition of endometritis. To answer this question, we should have a large number of cases that we do not currently have. The study started from a thorough in vitro simulation work with slaughterhouse material: we needed to acquire the perfect mastery of cytological smear techniques, the microscopic recognition of PMNs, and so on. To do this, we also used bovine and sheep material from young subjects and, therefore, with a high probability of not having endometritis conditions. Under these conditions, the method worked very well. Then, we moved to slaughtered mares preferring yearlings. Also, in this case, the technique worked well, although we immediately noticed that some ciliated cells exhibited a high fluorescence. Finally, we moved to live mares, recovering uterine cells from quick uterine flushing performed before insemination to carry out the cytological and microbiological diagnosis. Here, as reported in the paper, we found that the discrimination between clearly positive and negative samples on cytological examination was not as broad as we would have expected. Additionally, we encountered some false negatives. We believe that this occurrence is attributable to the condition of degeneration, autophagy and/or NETosis to which PMNs are subjected at the end of their activity. This condition is associated with a significant reduction in their ability to produce ROS. Considering that with simple flushing of the uterus without any transrectal manipulation (massage) it is possible to recover mainly epithelial flaking cells and inflammatory cells at the end of their activity, we think that the technique, although based on a solid finding, requires precautions and an optimization to increase its diagnostic power.
Furthermore, considering that fluorescence was performed only when turbidity of the samples applied the absorbance spectrum of 550 nm, would it be possible to also perform the examination using a lower or higher spectrum?
Reply. As reported in the text (Line 161), before adopting the absorbance spectrum of 550 nm, we examined various samples by scanning all the available absorbance spectra (between 380 and 700 nm) without detecting substantial differences. In addition, we found in the literature a study that used to examine the uterine flushing liquid of mares subjected to induced endometritis an assimilable value (545 nm); this comforted us. Therefore, we believe it is possible to use a different wavelength; however, it is crucial to use the same wavelength for the whole experiment, as in the case of microbiological, yeast and cell concentration densitometry.
The English language is appropriate, the few typset errors are mentioned and should be corrected.
Specific comments:
Introduction:
L44: „...poor perineal conformation, a uterine pending position...”
Reply. done
Material and Methods:
L110: „®” in upper case
Reply done
L119: „...involved 15 regularly slaughtered mares and 17 mares housed...”
Reply. done
L133: „...from slaughtered or living mares...”
Reply. done
L140: „…40 µm…”
Reply. done
L162: „…10 mL…”
Reply. done
L164: „…1 mL…”
Reply. done
L166: „…2.5 mL…5 mL…”
Reply. done
L167: „…2.5 mL…”
Reply. done
L172: „...4 µL...”
Reply. done
L200: „…490 nm…”
Reply. done
L218: „…Staphylococcus aureus...”
Reply. done
L251: „…procedure. The minimum…”
Reply. done
L269: „Single or combined…”
Reply. done
L271: „…used single or…”
Reply. done
L298: „…flushings, which…”
Reply. done
L328: in Table 1: „n” without the dot
Reply. done
Discussion:
L359: „...because it is easy to use...”
Reply. done
L401: „…of the collected uterine fluid….”
Reply. done
References:
The following references should be revised: 2, 5, 10, 14, 16, 17, 18, 24, 25, 26, 27, 28, 30, 35, 37, 41, 42.
Typical mistakes: abbreviation of some journal names while full names in others, misspelling, not using initial capital letters in journal names, occasionally missing endpages.
Reply Sorry, we used Endnote and I fear that in the various review steps between the authors the link between the Endnote file and the text is skipped. We revised the references, accordingly.
Reviewer 2 Report
The document titled Fluorescence spectroscopy for the diagnosis of endometritis in the mare” menciona las posibles implicaciones diagnosticas mediante el desarrollo de un nuevo método diagnostico a base fluorescence spectroscopy en endometritis in the mare. However, some adjustments are necessary before final publication.
Comments
Materials and methods
Line 236: the characteristics of the mares that underwent uterine lavage, age, number of births, etc. are not mentioned. As well as, if any examination was performed to determine the presence of clinical or subclinical endometritis in each animal. In this sense, was any mare identified with conformation problems or some other pathological condition that generated an inflammatory problem in the uterus?
Results
General: because only results of samples of uterine flushing which were positive and negative on the cytological test and Fluorescence spectra. But, why was a bacteriological examination of the sample or serological tests not considered?
Line 319-321. It is mentioned “we found seven false positives (53.8%) and three false negatives (11.5%) for the volume of the fluid collected at uterine flushing (cut-off level = 560 mL) whereas for the turbidity of such fluid (cut-off level = 0.025 Abs550) the number of false positives was two (15.4%) and the number of false negatives was two (11.5%).” In this document, how can this high percentage of errors be reduced? increase this part more.
Line 348-354. It would be convenient to reference this part based on what they mention
Discussion
I recommend a restructuring of this section based on the most important findings, mentioning why they occurred. For example, fluorescence spectrum variations, sample volume variations, turbidity, and Intracellular H2O2 content.
Please, correct the format of references; the name of journals must be written in abbreviation
Author Response
Reviewer 2
The document titled Fluorescence spectroscopy for the diagnosis of endometritis in the mare” menciona las posibles implicaciones diagnosticas mediante el desarrollo de un nuevo método diagnostico a base fluorescence spectroscopy en endometritis in the mare. However, some adjustments are necessary before final publication.
Comments
Materials and methods
Line 236: the characteristics of the mares that underwent uterine lavage, age, number of births, etc. are not mentioned. As well as, if any examination was performed to determine the presence of clinical or subclinical endometritis in each animal. In this sense, was any mare identified with conformation problems or some other pathological condition that generated an inflammatory problem in the uterus?
Reply. We thank the reviewer for the work done to improve the quality of our paper. As indicated in the paper, the mares were client-owned and housed on various private farms, and routinely subjected to uterine flushing for microbiological tests before undergoing artificial insemination. We added the breed, the mean ± SD of the age, and the number of births as well as the period in which the experiment was carried out. Furthermore, we stated that all the animals enrolled in this study did not have particular health or reproductive problems detected in the anamnesis or conformational problems that would require special attention (Lines 118-129).
Results
General: because only results of samples of uterine flushing which were positive and negative on the cytological test and Fluorescence spectra. But, why was a bacteriological examination of the sample or serological tests not considered?
Reply. As referred (Line 183), we used the traditional cytological assay as the gold standard and, hence, our results were related to this method. We did not conduct serological tests considering them outside the scope of the study. This could be a useful hint for a future study. Regarding the microbiological test, a detailed report of the microbiological examination has been added to the Results (Lines 318-321).
Line 319-321. It is mentioned “we found seven false positives (53.8%) and three false negatives (11.5%) for the volume of the fluid collected at uterine flushing (cut-off level = 560 mL) whereas for the turbidity of such fluid (cut-off level = 0.025 Abs550) the number of false positives was two (15.4%) and the number of false negatives was two (11.5%).” In this document, how can this high percentage of errors be reduced? increase this part more.
Reply. The assessment of the volume and the turbidity of the fluid collected from uterine flashing did not represent the focus of this study; these parameters were derived as collateral reliefs from the analysis conducted. On the basis of the statistically significant differences in these parameters between the two groups under comparison, it seemed only right to analyze these parameters as well as the fluorescence cytology under study. The volume of fluid recovered from uterine flushing, despite the differences between the groups under comparison, showed a large incidence of false positives such as not suggesting solid diagnostic reliability. Furthermore, the easy association between this parameter and the presence of fluid in the uterine lumen, which is easily detectable with ultrasound examination, made the parameter itself not very useful from a diagnostic point of view. On the other hand, turbidity of the uterine fluid proved to be a more reliable parameter with a reduced incidence of false positives. Unfortunately, apart from the study that we cited (Mattos et al., 1997), we did not find in the literature more studies on this parameter, which is very simple to evaluate and, therefore, very interesting to propose in clinical practice. As suggested, we have added in the conclusions a greater emphasis on this parameter (Lines 440-444).
Line 348-354. It would be convenient to reference this part based on what they mention
Reply. Unfortunately, to our knowledge, no experimental comparison has been made between fluorescence microscopy and spectroscopy, and flow cytometry techniques. Therefore, the comparison between these techniques can currently only be conducted on a speculative basis. Limiting factors and strengths of flow cytometry can be, however, always on the basis of a speculative analysis, sought in the paper that we have added to the references: Majka, S. M., Miller, H. L., Helm, K. M., Acosta, A. S., Childs, C. R., Kong, R., & Klemm, D. J. (2014). Analysis and isolation of adipocytes by flow cytometry. In Methods in enzymology (Vol. 537, pp. 281-296). Academic Press.
Discussion
I recommend a restructuring of this section based on the most important findings, mentioning why they occurred. For example, fluorescence spectrum variations, sample volume variations, turbidity, and Intracellular H2O2 content.
Reply. As suggested, we have reconsidered the discussion by giving in the conclusions a greater emphasis on the turbidity of the uterine fluid (Lines 440-444). Moreover, we did not detect significant fluorescence spectrum variations, and, as previously mentioned, the variations in the amount of fluid recovered are largely associated with the fluid already present in the uterine cavity. This aspect has been widely discussed. The content of H2O2 has also been extensively discussed; as a further source of variability of this parameter, NETosis has been introduced and supported by a new reference.
Please, correct the format of references; the name of journals must be written in abbreviation
Reply. Sorry, we used Endnote and we fear that in the various review steps between the authors the link between the Endnote file and the text is skipped. We revised the references, accordingly.
Round 2
Reviewer 2 Report
The manuscript improved considerably.
I suggest its publication in this form.
Author Response
Thanks a lot for your help
